# Unsupervised Multistep Deformable Registration of Remote Sensing Imagery Based on Deep Learning

**Maria Papadomanolaki** [1,2,*] **, Stergios Christodoulidis** [2] **, Konstantinos Karantzalos** [1] **and Maria Vakalopoulou** [2,3]

1   Remote Sensing Laboratory, National Technical University of Athens, 15780 Zographos, Greece; karank@central.ntua.gr
2   MICS Laboratory, CentraleSupélec, Université Paris-Saclay, 91190 Gif-sur-Yvette, France; stergios.christodoulidis@centralesupelec.fr (S.C.); maria.vakalopoulou@centralesupelec.fr (M.V.)
3   Inria Saclay, 91190 Gif-sur-Yvette, France
*   Correspondence: marpapadomanolaki@mail.ntua.gr

**Abstract:** Image registration is among the most popular and important problems of remote sensing. In this paper we propose a fully unsupervised, deep learning based multistep deformable registration scheme for aligning pairs of satellite imagery. The presented method is based on the expression power of deep fully convolutional networks, regressing directly the spatial gradients of the deformation and employing a 2D transformer layer to efficiently warp one image to the other, in an end-to-end fashion. The displacements are calculated with an iterative way, utilizing different time steps to refine and regress them. Our formulation can be integrated into any kind of fully convolutional architecture, providing at the same time fast inference performances. The developed methodology has been evaluated in two different datasets depicting urban and periurban areas; i.e., the very high-resolution dataset of the East Prefecture of Attica, Greece, as well as the high resolution ISPRS Ikonos dataset. Quantitative and qualitative results demonstrated the high potentials of our method.

**Keywords:** spatial gradients; deformation; satellite; very high resolution imagery; urban and periurban; learning-based registration; dense displacements; alignment



## 1. Introduction

The immense availability of earth observation data has enabled the remote sensing community to make great progress on a variety of applications. More often than not, these applications require that the employed images are on the same coordinate system, making the registration process a necessary prerequisite. For example, in change detection studies, images that depict the same region in different time stamps need to be properly registered, to reduce false positives that may occur due to the misalignment of the pair. Such a condition is sometimes quite difficult to be accomplished, as the available multitemporal images may have been collected from different aerial and satellite sensors, with different spatial resolution and acquisition angles. Moreover, there are cases that multitemporal data from the same family of satellites (e.g., Copernicus Sentinel-2A and Sentinel-2B) or from satellites with similar spatial and spectral characteristics (e.g., Landsat-8 and Sentinel-2) are not spatially aligned.

As a consequence, the registration process tackles a challenging problem especially in multitemporal datasets focusing on the elimination of geometric errors which are mainly caused by the different viewpoints of the aerial and satellite sensors. Moreover, since the majority of registration algorithms depend on image matching principles, various image classes with different intra-annual spectral behavior, dynamic classes like building shadows, illumination inconsistencies and general spectral variations hinder even more the accurate image alignment. These problems become even more challenging for high resolution data in complex urban environments that include local displacements on high structures such as buildings.

There are two main types of transformations that are usually employed for image registration: the linear or rigid [1–3] and the deformable or elastic [4–8]. The former deals with global transformations like rotation, translation and scale, while the latter focuses more on local geometric variations and pixelwise deformations. Moreover, in some works, more than one stage is used to estimate the transformation parameters of unregistered image pairs [9]. For example, in [10], the registration is performed in two stages; firstly, a block-weighted projective transformation model is used to extract feature points based on the SIFT algorithm. These points are used to estimate the projective transformation model and resample the moving image. After that, the outlier points are removed using an outlier-intensities model based on the Huber estimation. Similarly, in [11], an automated image registration process is proposed to handle the registration of hilly areas with many elevation variations. The altitude alterations hinder the successful image alignment as they introduce more complex local displacements. Firstly, the moving and reference images are decomposed into low level features. Then, some distinct feature point pairs are extracted, the relationship of which is calculated using feature-based matching methods. Least square matching is then employed to refine these initial points and finally obtain the transformation coefficients. In recent years, the advances of deep learning have led the computer vision community to registration solutions that are more related to neural networks, especially in the medical field [12–15]. Despite this progress, little effort has been made to adjust these frameworks for image registration in remote sensing, with semiautomated algorithms still being widely employed [1,16,17]. However, such methods do not provide end-to-end mapping functions while they are usually inefficient in real time applications due to their computational complexity especially if we consider the high dimensionality of the remote sensing imagery.

Starting with methods that focus on the use of deep learning for image matching, in [18], the authors proposed a supervised way to perform patch level matching for remote sensing imagery. In particular, patches centered at key points in the moving and reference image are used. A deep network then receives as input the pairs and decides if these patches are corresponding or not. After defining the matching pairs, least squares are used to define the best affine transformation parameters. Furthermore, Ref. [19] exploits the convolutional layers of a pretrained VGG-16 network [20] to formulate multiscale feature descriptors [21] for efficient matching of points. The extracted features are then compared based on distance metrics while a dynamic inlier point selection is also proposed. Additionally, Ref. [22] employs a siamese neural network to approximate reliable matching pairs between multitemporal images. Similar approaches are also being proposed to discover proper matching pairs on multisource settings, like for example in [23] where a correspondence heatmap is generated through a multiscale, feature-space cross-correlation operator to identify matching pairs of optical and SAR images. Moreover, generative networks are also utilized in [24] for the same purposes.

The literature is more sparse, however, when it comes to deep learning algorithms focusing directly on the registration between pairs of images. Authors in [25] exploit a DenseNet [26] to approximate directly a rigid transformation matrix of a given image pair by regressing the four corner displacements that align in the best way the reference and the moving image. In [27], the elastic registration problem is tackled by a fully convolutional scheme which calculates the deformation and learns scale-specific features in order to align optical images to cadastral maps. The same method was improved in [28] formulating a multitask framework which integrates the task of semantic segmentation on buildings for cadastral maps and optical imaging.

In this paper, we build upon the recent work of [29], where an end-to-end training of a deep learning model is employed coupling both linear and deformable registration problems focusing on medical imaging. A similar setup is also presented in [30] for remote sensing imagery, reporting very competent performances on the alignment of very high resolution satellite imagery. Here, we extend the previous formulations by introducing a multistep approach for learning an accurate mapping between unregistered image pairs.

More specifically, the contributions of our paper are three fold: (i) we present a multistep deformable image registration approach, which aligns the given image pairs through an iterative process refining at each step the deformation that has been obtained in the previous step, (ii) an end-to-end fully unsupervised formulation for the registration of optical remote sensing data and (iii) a modular framework that can be adapted to any network and loss function, opening a lot of potential for a variety of remote sensing applications. Our models and code can be found at https://github.com/mpapadomanolaki/Multi-Step-Deformable-Registration (accessed on 14 March 2021).

The rest of the paper is organized as follows: In Section 2, we explain the proposed methodology in detail. In Section 3 we describe the employed datasets and we provide quantitative and qualitative experimental results. Then, in Section 4 we make discussion comments and finally in Section 5 we end with a conclusion.

## 2. Materials and Methods

### 2.1. Multistep Deformable Registration Network

During the image registration process, the aim is to align two or more images and bring them to the same coordinate system. More specifically, without loss of generality let us define a pair of images that we want to register as source ($S$) and target image ($R$). The goal of deformable registration is to obtain a transformation map $G$ which aligns in the best way the $S$ to the coordinate system of $R$ and calculate the registered image $D$ using the following operation

$$D = W(S, G),$$

where $W(., G)$ is a sampling operation that warps the source image $S$ using an interpolation sampler $W$ according to the deformation $G$. This sampling operation firstly requires that the transformation parameters or displacement maps are estimated [31]. Inspired by [29], these parameters and/or transformation maps can be learned directly via a deep learning architecture.

In this paper, we propose a formulation for learning deformable, pixelwise displacements between a pair of images. In our framework, these displacements are considered as the spatial gradients $\Phi$ along the $d$ dimension, with $d \in \{x, y\}$, that is, the displacement fields for every pixel location. To ensure smooth displacements and avoid self-crossings on horizontal and vertical directions [32], the predicted spatial gradients $\Phi$ are constrained to be positive using a logistic function $L$ of the form

$$L([\Phi]_d) = \frac{c}{1 + (c - 1) \cdot e^{-[\Phi]_d}}$$

This restriction enforces the spatial gradients $[\Phi]_d$ to be in the range of $[0, c]$, where $c$ is a tunable parameter that represents the maximum displacement along consecutive pixels. Using this formulation for consecutive pixel locations $p$ and $q$, when $[\Phi(p)]_d < 0$, $L([\Phi(p)]_d)$ brings the pixels $p, q$ closer to each other on the dimension $d$, while when $[\Phi(p)]_d > 0$, the $p, q$ pixels are moved away from each other along the $d$ dimension. Finally, when $[\Phi_{Id}(p)]_d = 0$ then $L([\Phi(p)]_d) = 1$ (identity deformation), which means that neighboring pixels remain in the same position. Having determined the spatial gradients that we obtain and learn automatically from the network, the deformation grid per dimension $[G]_d$ can be obtained by calculating the cumulative sum along each dimension $d$. Then, the deformed image is produced using a 2D transformer layer that performs a backward bilinear interpolation. During this backward process, the spectral intensities for the displaced pixels in the warped image are generated using the original pixel coordinates of the source image $S$

$$D(p) = W(S, [G]_d)(p) = \sum_q S(q) \prod_d max(0, c - |[G(p)]_d - q_d|),$$

In order to enhance the model's performance and refine the calculated spatial gradients, we define the registration task as a recurrent process, outlined in Figure 1. Giving some more details, if we have a fully convolutional network, then the final spatial gradients $[\Phi_T]_d$ can be computed iteratively using the same network $t$ times, where $t \in \{1, \ldots, T\}$

$$[\Phi_T]_d = [\Phi_{Id}]_d + \sum_{t=1}^{T} [\Phi_t]_d$$

In the above equation, $\Phi_{Id}$ is the identity deformation. At each time step $t$, the spatial gradients of the previous time step $t-1$ are added to the spatial gradients of the current time step $t$, resulting in the deformation grid $[G_t]_d$ forming each time the deformed image as

$$D_t = W(D_{t-1}, [G_t]_d),$$

In every iteration $t$, the target image $R$ is compared with the warped source image $D_{t-1}$ under the deformation $[G_t]_d$.

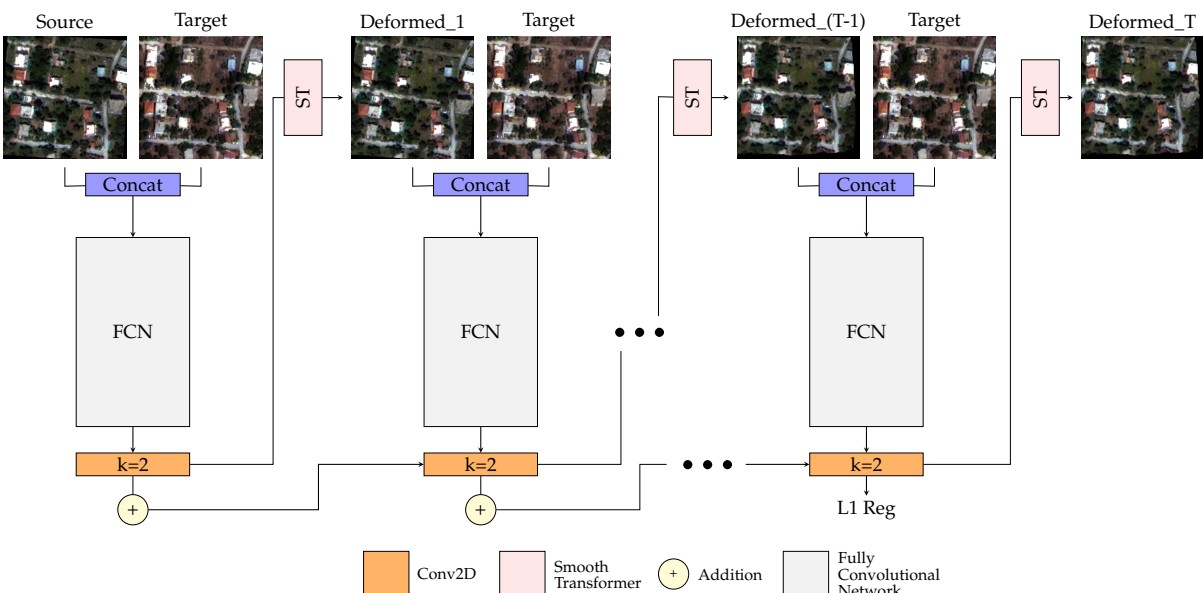

**Figure 1.** The proposed multistep deformable registration approach. At every time step $t$, the current transformation parameters are added to the transformation parameters of the previous time step $t-1$. Through this iterative process, the network learns to properly refine the initial spatial gradients, providing more accurate deformation grids.

Our proposed framework is completely unsupervised in the sense that the training does not require the use of pairs with known deformations, something that is very difficult to obtain in practice. In particular, the training of our method is performed by calculating the similarity between the spectral values of the deformed image $D$ and the target image $R$. In this work, we employ the mean square error (MSE) loss; however, many other loss functions can be utilized, such as the Normalized Cross Correlation (NCC), Local Cross Correlation (LCC) [33] or mutual information [34]. Moreover, in order to ensure that the spatial gradients are smooth and to prevent very noisy deformations, we employ an additional regularization to the spatial gradients enforcing them to be very close to $[\Phi_{Id}]_d$. Formally our loss function is defined as

$$\mathbf{L} = \frac{1}{T} \sum_{t=1}^{T} ||R - W(D_{t-1}, [G_t]_d)||^2 + \beta || \sum_{t=1}^{T} ||[\Phi_T]_d - [\Phi_{Id}]_d||$$

where $\beta$ is a regularization weight. Large $\beta$ values result in deformations that are closer to the identity, which means that displacements remain smooth and not exceptionally large. On the contrary, smaller $\beta$ values widen the space of the transformation but at the risk of

noisy deformations. As $D_{t-1}$ we denote the $S$ image at each time step. One can observe that for the final loss the MSE is calculated using all the deformed images $D_t$ for $t \in \{1, \dots, T\}$, while the regularization term is only applied on the final obtained spatial gradients $[\Phi_T]_d$.

### 2.2. Network Architecture

Our framework is modular and it does not depend on the proposed deep architecture. For this study, we chose to explore an autoencoder which downsamples the input through the encoder and then upsamples it back through the decoder. The overview of the employed fully convolutional architecture is summarized in Figure 2. The source image $S$, or the deformed image $D_t$ produced in each time step, and the target image $R$ are firstly concatenated along the channel dimension before being forwarded through the network. The encoder comprises of four convolutional layers, each of which involves a single strided convolution operation with a receptive field equal to 3, as well as instance normalization and a LeakyReLU activation function. Beginning from the first layer, the channels increase from 16, to 32, 64 and 128, while at every layer except the fourth, the dimensions of the input are reduced in half using a max pooling layer. After the last encoding layer, a decoding branch receives the downsampled image volume and restores it back to its original dimensions through 4 convolutional layers. Symmetrical to the encoder, all decoding layers except the first one include an upsampling operation, while all layers involve a single strided convolution operation with a receptive field equal to 3, as well as instance normalization and a LeakyReLU activation function. Beginning from the first decoding layer, the channels are restored back from 128, to 64, 32 and 16. After the decoding branch, a single strided convolutional layer with a receptive field equal to 3 reduces the channel dimension from 16 to 2 with each channel providing the spatial gradients per dimension. The deformation grids $[G]_d$ are obtained by applying the cumulative operation and are then inserted to the spatial transformer layer. The output of the network is the deformed image $D_t$ together with the displacement grids per dimension.

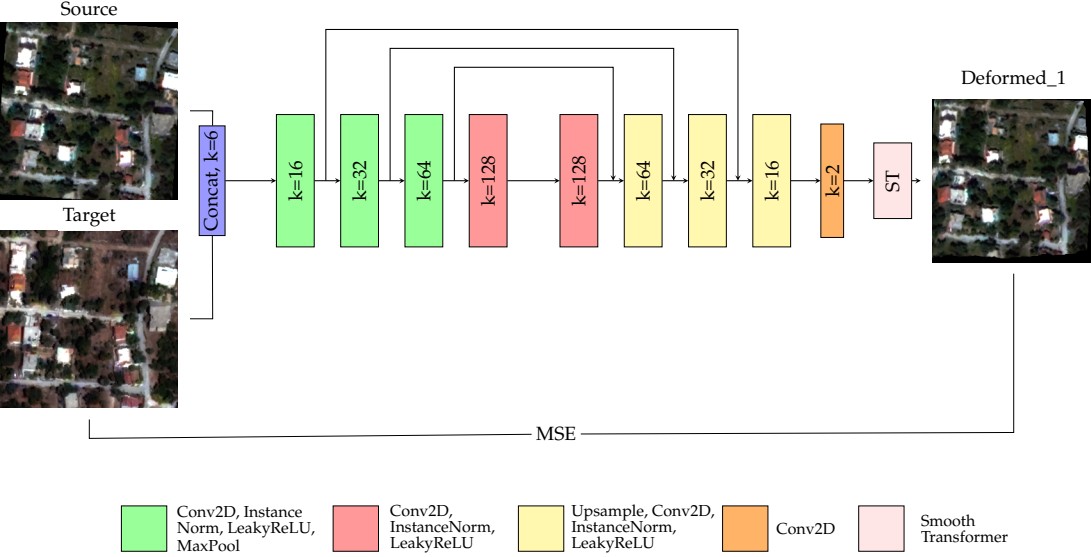

**Figure 2.** Overview of the proposed fully convolutional network employed during the first time step. The concatenation of the source image $S$ and the target image $R$ is given as input to the network.

## 3. Experimental Results

In this section we present an extensive quantitative and qualitative evaluation of our method and compare it with other methods existing in the literature. We also perform an ablation study to discover the influence of the time step $t$ on the registration process. The evaluation of the method is performed by calculating the errors on landmark location points extracted from the testing images. In particular, 15 landmark points were chosen

separately for building areas and for areas that depict roads and fields. In all the cases, corner points were mainly selected, corresponding to building roof corners or corners of field borders as well as road corners and crossroads. Reporting separately the errors on buildings and roads or fields was performed in order to further confirm the effectiveness of the proposed framework, since building areas are more challenging due to the parallax errors caused by the different viewpoints of the sensors.

### 3.1. Datasets

Our framework has been extensively evaluated on two different very high resolution datasets which depict some urban and periurban regions. In the following two subsections we describe these datasets in detail.

#### 3.1.1. Attica VHR

The first dataset is comprised of two very high resolution images that depict a 9 km$^2$ region in the East Prefecture of Attica, Greece, in 2010 and 2011. The images are both acquired by the WorldView-2 satellite and their dimensions are approximately 8000 by 7000, with every sample having been pansharpened and atmospherically corrected. This dataset contains urban and periurban areas with flat and hilly regions. For our investigation, the whole region was divided into 36 equal nonoverlapping subregions of approximate size 1100 by 1300 pixels; 20 of them were used for training, 6 for validation and 10 for testing. For the training process, patches of size $256 \times 256$ were produced with a stride of 128, resulting in 840 training, 120 validation and 200 testing samples.

#### 3.1.2. ISPRS Ikonos

For further evaluation, we employed the ISPRS Ikonos (https://www.isprs.org/data/ikonos/default.aspx (accessed on 14 March 2021)) dataset that provides one bitemporal pair of high resolution images. As the name of the dataset suggests, the images are acquired by the Ikonos satellite and their dimensions are 2000 by 2000 pixels. They depict a 2 km$^2$ region and they have one panchromatic channel available. This dataset depicts a flat region with high buildings. 3/4 of the images were used for training and validation, while the rest was intended for testing purposes. Specifically, patches of size $192 \times 192$ were extracted with a step of 80, forming 400 training, 40 validation and 60 testing samples.

### 3.2. Optimization

Hyperparameters were similar for both datasets, picking Adam optimizer with a learning rate of $10^{-4}$ and batch size equal to 1. The models were trained for 100 epochs, while early stopping was used for the selection of the best model, taking into account the training and validation performances. In addition, random deformed images from the validation part were visualized to verify the registration results. For our experiments we used $t = 3$ time steps, which gave us a good trade off between computational complexity and performance. The regularizer $\beta$ was set to to $10^{-6}$ for both datasets after a thorough grid search was conducted so that the optimal results could be obtained. In addition, the parameter $c$ for the logistic function $L$ was equal to 2 and 8 pixels for the Attica VHR dataset and the ISPRS IKONOS dataset respectively. All experiments were implemented using the PyTorch deep learning library [35] on a single NVIDIA GeForce GTX TITAN with 12 GB of GPU memory.

### 3.3. Ablation Study

The influence of the time step $t$ was investigated by conducting an ablation study, examining the performance of the proposed multistep method using different time steps. The experiments were carried out on the Attica VHR dataset. More specifically, we trained the proposed formulation using $T = \{1, 2, 3, 4, 5\}$ time steps, while for every different experiment we validated the performances on the testing landmark location points. The results are demonstrated in Table 1 for building areas as well as for areas that include roads

and fields. $[\Phi]_d$ ($T = 1$) is essentially the method presented in [30] using the setup and parameters proposed on this study. As one can notice from Table 1, the one step approach registers the pair of images; however, it reports higher errors than the methods that use our proposed iterative formulation. This is the case both for buildings as well as roads and fields. As far as the building areas are concerned, the errors are ameliorated for *dy* and *ds* axes when 2 steps are employed, while in the case of 3 steps, all axis errors are much improved compared to the 1 step approach. The *ds* error is further boosted when $T = 4$, and lastly, when $T = 5$ the sum of all axes errors is minimized. The landmark location errors concerning areas that depict roads and fields follow a similar behavior as the steps rise. That is, all axis errors are improved when $T = 2$, falling below 1 pixel when $T = 3$. Specifically, when $T >= 3$, all axis errors become lower than 1 pixel, while the sum of all axes errors is minimized when $T = 5$. We should mention at this point that we also conducted experiments using more than $T = 5$ time steps; however, not any further improvement was noticed. Nevertheless, we can not say that $T = 5$ is the limit for the proposed framework. That is because this multistep scheme is independent of the employed deep architecture, which means that adapting a different network and applying more time steps could lead to even better results for a different type of application.

**Table 1.** Results from the ablation study on the testing part of Attica VHR dataset. Landmark location errors are calculated for deformable registration models that are trained with various time steps (*T*). The last column indicates the time needed by each framework to complete one training epoch.

| Method | Buildings | | | Roads and Fields | | | Training Time per Epoch (sec) |
|---|---|---|---|---|---|---|---|
| | dx | dy | ds | dx | dy | ds | |
| Unregistered | 10.73 | 7.60 | 13.53 | 8.20 | 7.53 | 11.31 | - |
| $[\Phi]_d$ ($T = 1$) | 2.40 | 2.33 | 3.67 | 1.60 | 1.67 | 2.59 | ≈22 |
| $[\Phi]_d$ ($T = 2$) | 3.07 | 0.60 | 3.40 | 0.80 | 1.20 | 1.75 | ≈28 |
| $[\Phi]_d$ ($T = 3$) | 2.00 | 1.20 | 2.60 | 0.47 | 0.53 | 0.91 | ≈34 |
| $[\Phi]_d$ ($T = 4$) | 2.13 | 1.00 | 2.55 | 0.20 | 0.80 | 0.90 | ≈44 |
| $[\Phi]_d$ ($T = 5$) | 1.27 | 1.27 | 2.07 | 0.33 | 0.33 | 0.59 | ≈56 |

For this ablation study, we trained our formulation different times changing only the number of time steps (*T*) and keeping the rest of the employed hyperparameters the same. Based on Table 1 we can draw the conclusion that under this specific framework, the results become optimal when 3 or more steps are applied. In our case, the $T = 3$ choice seems the most favorable, since it produces much better errors compared to the noniterative method ($T = 1$). Beside the registration errors, computational time is an additional aspect that someone should consider for the selection of the best *t* variable. In particular, as it is demonstrated in Table 1, the time needed by the model to complete one training epoch is getting larger with the additional iteration steps. Again, $T = 3$ seems to give a good trade off between accuracy and time complexity. For these reasons, in the following experimental results, the quantitative and qualitative evaluation of our method is conducted using $T = 3$ time steps.

### 3.4. Quantitative and Qualitative Evaluation

Our method has been evaluated both quantitatively and qualitatively on two different datasets and it has been also compared with other registration frameworks from the literature: the method proposed in [30] as well as the greedy fluid flow algorithm [36]. The method in [30] employs a deep learning based pipeline, where the affine parameters and the deformable deformations are directly predicted by the pipeline itself. On the contrary, the method in [36] is not a learning based method, with the transformation parameters being calculated by integrating velocity fields forward in time, using the squared error dissimilarity metric on the space of diffeomorphisms. The optimization of the problem

is conducted in an iterative way, where at every iteration the current transformation parameters are approximated based on the previous transformation parameters and the previous velocity field. For this implementation we used the greedy registration library [37]. As the method suggests, a deformable registration is applied on the given image pair, after having firstly been preprocessed with an affine transformation step. For each of these methods we evaluate separately the different registration components and refer to them as "only $A$", which is the plain affine registration, "only $\Phi$", which is the plain deformable registration and "$A \,\&\, \Phi$", which is the combination of both. For the quantitative evaluation, we compute errors measured as average euclidean distances for landmark locations on the testing images (Tables 2 and 3). For the qualitative evaluation, we provide different pairs of images from the testing regions, using checkerboard visualizations. The visualizations concern the target image $R$ and the warped source image $D$ before and after the registration (Figures 3–5). It should be mentioned here that for the Attica VHR dataset, we added further displacements on the image pairs compared to [30], e.g., rotation, translation and scale, in order to further complex the exploited deformations and make the problem more challenging. Specifically, the extra translation ranges from 2 to 12 pixels, the rotation from 3 to 4 degrees, while the scale factor is equal to 2 or 3.

**Table 2.** Errors measured as average euclidean distances between 15 landmark locations on the testing images of the Attica VHR dataset. $dx$ and $dy$ represent the distances along the $x$ and $y$ axis, respectively. $ds$ indicates the average error along all axes per pixel, while the last column indicates the time needed by the methods to register a pair of $256 \times 256$ images. Errors on building landmark locations (**left**) and errors related to points on roads and fields (**right**) are presented.

| Method | Buildings | | | Roads and Fields | | | Inference Time (sec) |
|---|---|---|---|---|---|---|---|
| | **dx** | **dy** | **ds** | **dx** | **dy** | **ds** | |
| **Unregistered** | **10.73** | **7.60** | **13.53** | **8.20** | **7.53** | **11.31** | **-** |
| only $A$ [30] | 3.60 | 2.47 | 4.73 | 1.93 | 1.93 | 2.96 | ≈0.009 |
| only $\Phi$ [30] | 2.40 | 2.33 | 3.67 | 1.60 | 1.67 | 2.59 | ≈0.005 |
| $A \,\&\, \Phi$ [30] | 2.00 | 2.53 | 3.30 | 0.73 | 1.33 | 1.77 | ≈0.006 |
| only $A$ [36] | 2.00 | 2.33 | 3.30 | 0.87 | 1.07 | 1.53 | ≈0.489 |
| only $\Phi$ [36] | 4.20 | 3.53 | 5.89 | 2.80 | 3.00 | 4.50 | ≈0.664 |
| $A \,\&\, \Phi$ [36] | 1.80 | 2.20 | 3.13 | 0.93 | 1.33 | 1.88 | ≈1.284 |
| *Proposed* | 2.00 | 1.20 | 2.60 | 0.47 | 0.53 | 0.91 | ≈0.012 |

**Table 3.** Quantitative evaluation on the ISPRS Ikonos dataset. Errors measured as average euclidean distances between 15 landmark locations. $dx$ and $dy$ represent the distances along the $x$ and $y$ axis, respectively. $ds$ indicates the average error along all axes per pixel, while the last column indicates the time needed by the methods to register a pair of $192 \times 192$ images. Errors on building landmark locations (**left**) and errors related to points on roads and fields (**right**) are presented.

| Method | Buildings | | | Roads and Fields | | | Inference Time (sec) |
|---|---|---|---|---|---|---|---|
| | **dx** | **dy** | **ds** | **dx** | **dy** | **ds** | |
| **Unregistered** | **2.93** | **9.60** | **10.09** | **2.47** | **9.13** | **9.53** | **-** |
| only $A$ [30] | 1.07 | 3.93 | 4.13 | 0.93 | 2.33 | 2.65 | ≈0.006 |
| only $\Phi$ [30] | 0.60 | 1.87 | 2.08 | 0.53 | 1.87 | 2.09 | ≈0.004 |
| $A \,\&\, \Phi$ [30] | 0.27 | 1.53 | 1.64 | 0.40 | 1.80 | 1.89 | ≈0.005 |
| only $A$ [36] | 0.60 | 1.47 | 1.75 | 0.40 | 1.60 | 1.76 | ≈0.378 |
| only $\Phi$ [36] | 2.06 | 3.20 | 3.95 | 1.87 | 4.00 | 4.65 | ≈0.583 |
| $A \,\&\, \Phi$ [36] | 0.53 | 1.27 | 1.55 | 0.87 | 1.20 | 1.73 | ≈1.102 |
| *Proposed* | 0.20 | 1.20 | 1.32 | 0.40 | 0.93 | 1.22 | ≈0.011 |

Unregistered            only *A*[30]            onlyΦ[30]            *A*&Φ[30]            *Proposed*

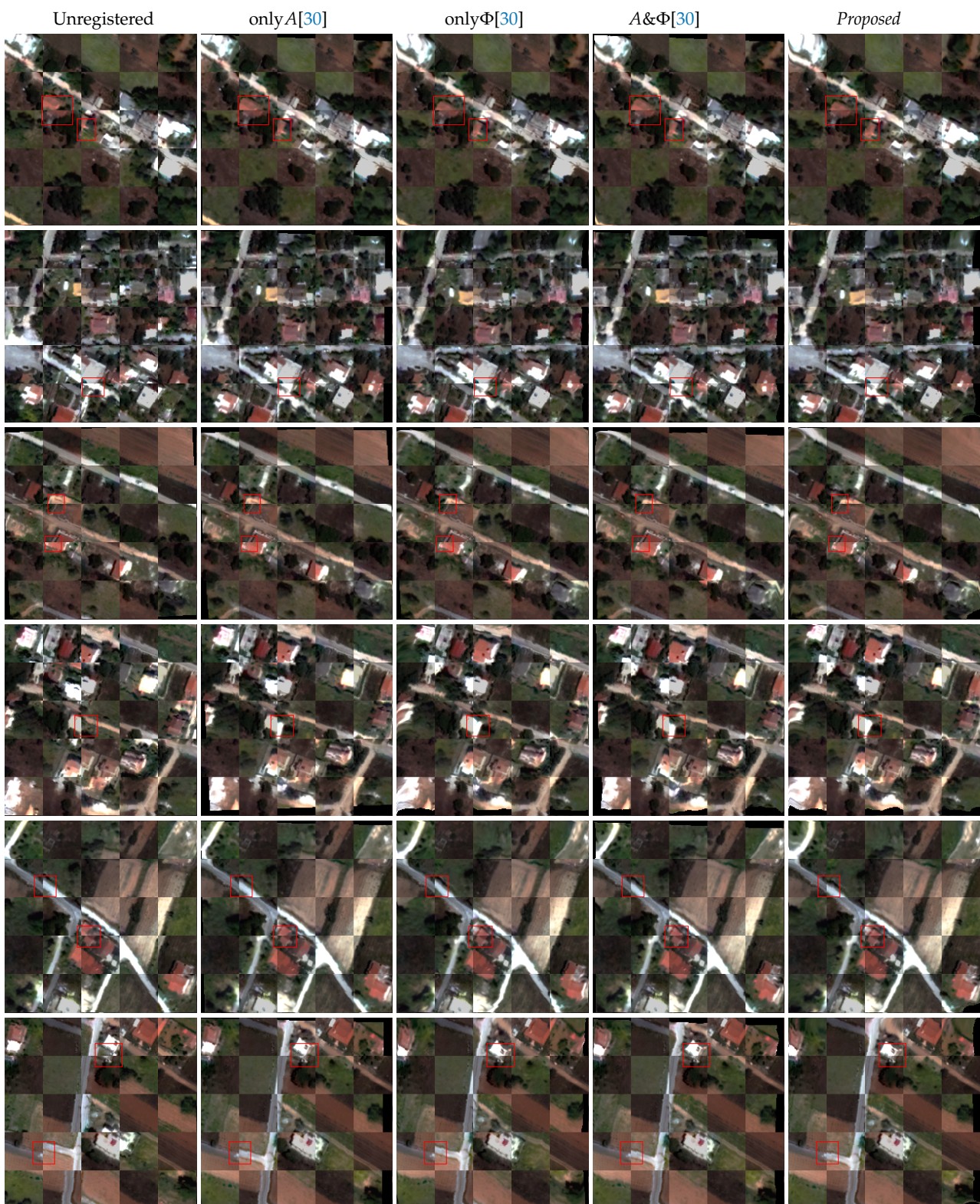

**Figure 3.** Qualitative evaluation on different pairs from the Attica VHR testing regions. Different approaches from [30] are compared i.e., from left to right: Unregistered raw data, results: only *A*, only Φ, *A* & Φ, *Proposed*.

| Unregistered | only $A$ [36] | only $\Phi$ [36] | $A$ & $\Phi$ [36] | *Proposed* |

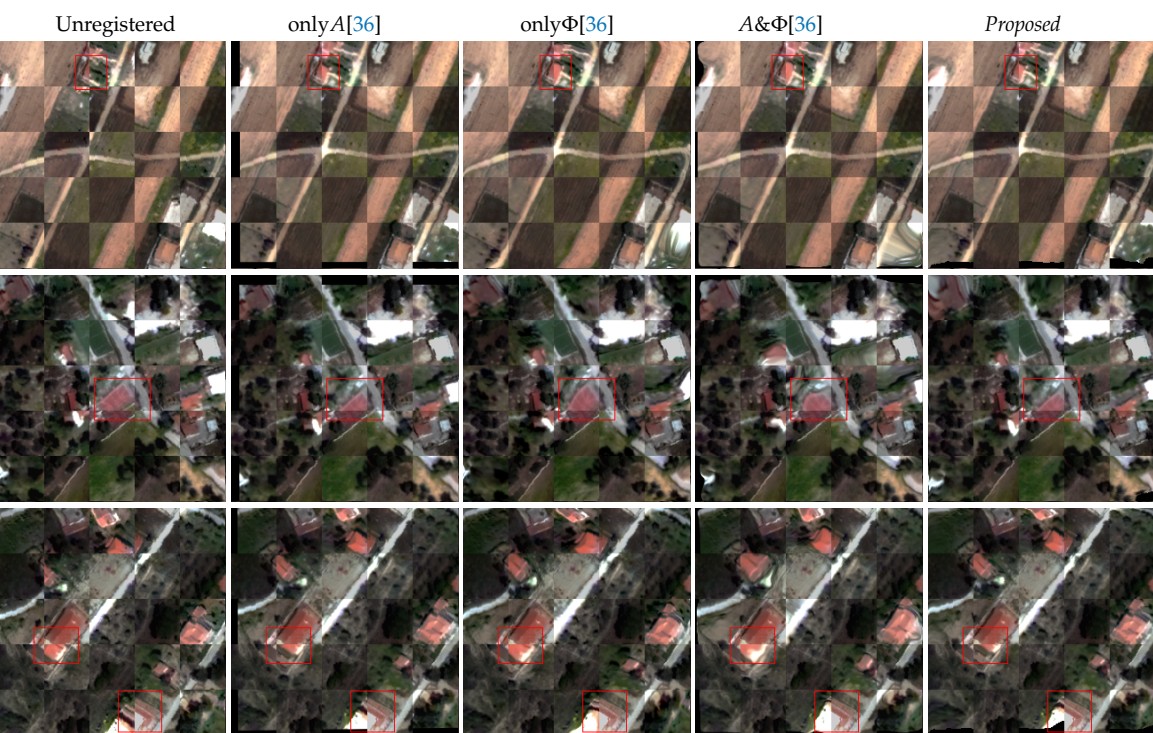

**Figure 4.** Qualitative evaluation on different pairs from the Attica VHR testing regions. Different approaches from [36] are compared i.e., from left to right: Unregistered raw data, results: only $A$, only $\Phi$, $A$ & $\Phi$, *Proposed*.

### 3.4.1. Attica VHR Dataset

Starting with the Attica VHR dataset and the method in [30], we can observe in Table 2 that the proposed method outperforms the rest of the approaches both in building areas as well as in areas that include roads and fields. Giving some more details, the largest errors on the building landmark location points have occurred in the only $A$ case, for all axes. As one can observe, the $ds$ error is almost 5 pixels, which shows that even though the affine transformation has managed to bring the two image pairs closer, it still misses some local information, leading to inconsistencies in the matching process. In the only $\Phi$ case, the $dx$ and $ds$ errors are reduced by 1 pixel, while the $dy$ error is slightly improved. When $A$ & $\Phi$ are combined, the synergy among the two different types of transformations enhances the registration performance on $dx$ and $ds$, while the $dy$ error is not improved. As far as the proposed method is concerned, the $dx$ error is the same as in the $A$ & $\Phi$ case, with the rest of the errors having been significantly ameliorated, attaining the lowest values compared to the rest of the methods. Continuing with the areas that include roads and fields, one can notice that the error for the unregistered image pair is smaller compared to the building areas. As we have already mentioned, this is due to the fact that these landmark location points are on the earth's surface, which means that their position is not so sensitive to the different viewpoints of the sensors. Here, all methods have managed to register these points with an error smaller than 3 pixels. Similarly to the building points, the results of $A$ & $\Phi$ are mitigated compared with only $A$ and only $\Phi$. The proposed method brings all the errors below 1 pixel, proving the efficiency of the multistep registration approach, which results in more aligned image pairs through the repeated refinement process. Continuing with the method in [36], one can observe in Table 2 that the best results are attained from only $A$ as well as $A$ & $\Phi$. The results for these two approaches are very similar, meaning that the deformable component in this case does not contribute much to the amelioration of the results. This is also proved by the only $\Phi$ results, which have resulted in high errors both for buildings as well as for roads and fields. This finding confirms that the task of registration in remote sensing can be very challenging due to the dense and complex geometry patterns presented in the images. It also highlights the

need to apply first a rigid transformation to bring the pair of images closer and then more complex dense deformations. Our method maximizes the potentials of the deformable registration without the need of any prerequisites, requiring also less time when it comes to inference, compared with [36]. The fast inference is very important for remote sensing registration if one considers the large size of earth observation images.

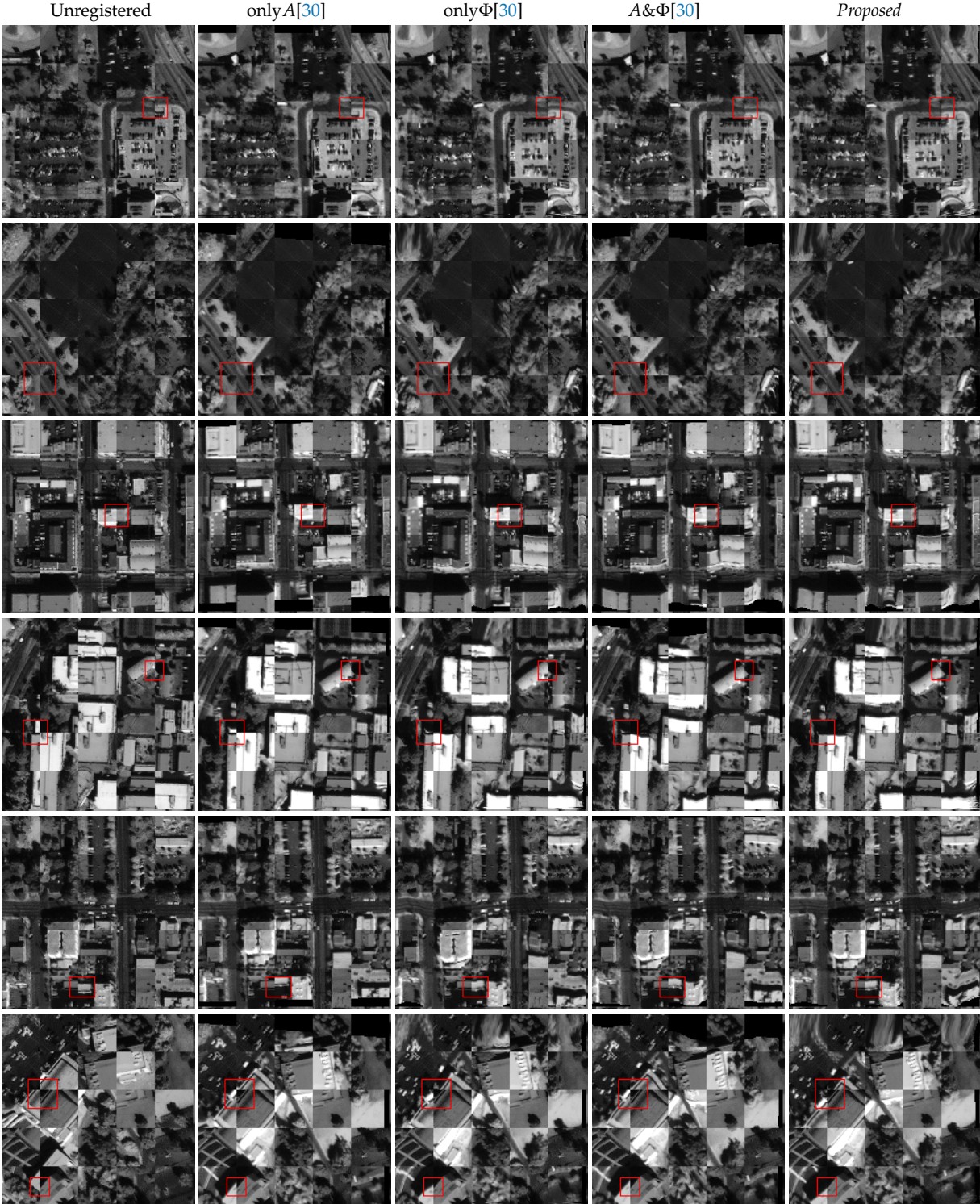

**Figure 5.** Qualitative evaluation on different pairs from the ISPRS Ikonos testing regions. Different approaches from [30] are compared, i.e., from left to right: Unregistered raw data, results: only *A*, only Φ, *A* & Φ, *Proposed*.

Visualized examples on the testing part of the Attica VHR dataset can be observed in Figures 3 and 4 for methods [30,36] respectively. Beginning with Figure 3, we can observe at the first row that the registration of the two highlighted buildings is much more consistent in the proposed approach compared with the rest of the methods. Furthermore, in the third row, the highlighted building and the field have failed to be properly registered in the case of only $A$ and only $\Phi$, while they are much more aligned in the case of $A$ & $\Phi$ as well as in our multistep approach. A similar example can also be observed in the fifth row for a building and a road. Additional visualized examples can be noticed in the rest of the rows. Continuing with Figure 4 of method [36], in the first row we have an example where the highlighted building has been more properly aligned by the proposed method. In the second row, the highlighted field has been aligned successfully by the proposed method as well as by the only $A$ approach. When the deformable component is integrated, however, the shape of this object is not preserved, with the boundaries having been significantly altered. Lastly, in the third row the proposed method managed to register more correctly the highlighted buildings as opposed to the rest of the approaches.

### 3.4.2. ISPRS Ikonos Dataset

Landmark location errors are also provided in Table 3 for the ISPRS Ikonos dataset. Here, the displacements of the unregistered image pairs are lower than the Attica VHR dataset, especially in the $x$ axis. However, what is challenging in this dataset is that the patches depict larger and higher buildings, meaning that different viewpoints of the sensors alter significantly the appearance of the buildings. Once again, for this evaluation we provide errors separately for building areas and for areas that depict roads and fields. Beginning with the method in [30] on the building areas, we can notice that the largest errors occur in the only $A$ case, while all errors are improved in the only $\Phi$ case. The combination of $A$ & $\Phi$ enhances further the model's performance since the errors are even lower in all axes, which means that the integration of the affine transformation contributed to a more constructive registration process. Lastly, the proposed method produced the most successful results in all axes. In the right part of Table 3, we can see the corresponding landmark location errors for the areas that include roads and fields. Here, the only $A$ method results in much better errors compared to the building outcomes, which is normal since roads and fields are not so much affected by the viewpoints of the sensors. Other than that, the rest of the methods follow a pattern similar to the building case, with the proposed method achieving the optimal results. Specifically, the multistep approach brought the $dx$ and $dy$ errors below 1 pixel, while the $ds$ error is also very close to 1 pixel. Additionally, in Table 3 the results for method [36] are provided. Here, the highest errors are attained by the only $\Phi$ approach. In the only $A$ case the errors are much ameliorated while they are slightly improved in the $A$ & $\Phi$ case where the affine transformation is applied as a preprocessing step. The $A$ & $\Phi$ errors are similar with our proposed method, although the sum of all axis errors is lower when our method is applied. Specifically, as far as buildings are concerned, the sum of axis errors is 3.35 for $A$ & $\Phi$ [36] and 2.72 for the proposed scheme. Regarding roads and fields, the sum of axis errors is 3.80 in the case of $A$ & $\Phi$ [36] and 2.55 in the proposed framework. Lastly, like in the Attica VHR dataset, our approach provides better inference times.

Some checkerboard visualizations from the testing part of the ISPRS Ikonos dataset are provided in Figure 5, highlighting the advantages of our method. Examples from building areas as well as areas with roads and fields are demonstrated, proving the better image alignment that our method has achieved. One issue that we can notice is that sometimes the methods that involve the deformable component result in curvy building boundaries. This deteriorated smoothness is inextricably related to the employed regularization term. For each method, we experimented with a wide range of regularization weight values, until the optimal result could be achieved. According to our experiments, the only $\Phi$ method was more difficult to control, while it was more easy to handle this issue when the affine component was integrated. The proposed multistep approach was the most convenient

method to fine tune, resulting in the most smooth alignments but not eliminating completely the curvy parts. It should be mentioned here that this problem was not so difficult to control in the Attica VHR dataset case in which the buildings are smaller and not so high. In addition, the data samples were much more rich, since we had almost 800 patches for training, compared to the Ikonos dataset where we had only 400. This smoothness issue is something that we want to further explore in the future, in an attempt to make the registration process even more robust.

In Figure 6 we can also observe the training-validation loss curves for all the employed methods. The only $A$ network produces the most noisy validation loss numbers, while after the 80th epoch there are even larger fluctuations and the losses do not further improve. In the $A$ & $\Phi$ case, we notice that a less noisy graph has been produced. In addition, the network seems to learn until the last epoch since the training and validation losses are decreasing. Regarding the proposed framework as well as the only $\Phi$ approach, they result in more smooth loss curves. Here, there are again many oscillations in the validation loss range; however, they are more limited compared to the methods that include the affine component and closer to the training loss values.

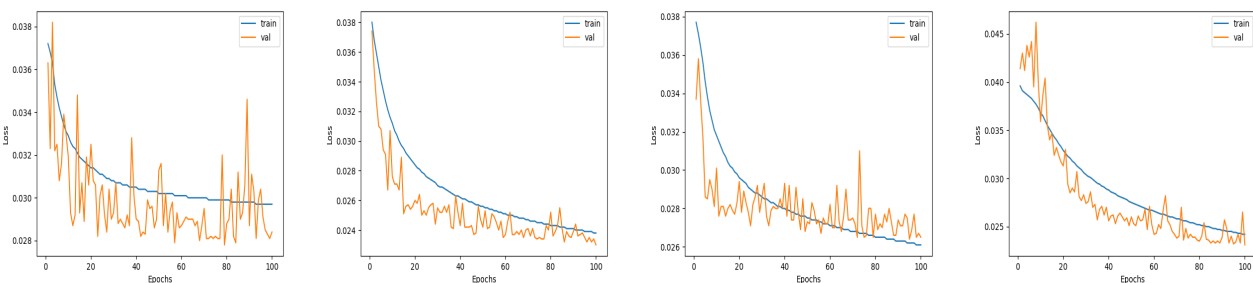

**Figure 6.** Training and validation loss curves for the different methods on the ISPRS Ikonos dataset. From left to right: only $A$, only $\Phi$, $A$ & $\Phi$, *Proposed*.

## 4. Discussion

Taking into account both the quantitative and the qualitative results, we can make some overall comments for the investigated methods. In general, when it comes to deep learning based approaches, the plain affine transformation produces the largest errors especially when building areas are included. This happens because such a transformation is global in nature and thus unable to handle properly the local displacements. When the deformable transformation is applied, the local displacements are handled in a better way with the results being much ameliorated, especially for buildings which are more prone to inconsistencies between the different dates. On the contrary, for methods that are not learning based, we found that the largest errors resulted from the plain deformable registration. In both types of methods, the combination of the affine and deformable registration ameliorates the results but not to a great extent. Both building areas as well as roads and fields are aligned more correctly when our multistep approach is employed, which leads us to the conclusion that the iterative registration process enables the model to refine the results and adjust the deformations more appropriately.

Compared with previous state-of-the-art approaches, the proposed scheme includes some novel features which can contribute to the further improvement of registration techniques concerning the remote sensing domain. Firstly, our method is able to generate dense deformations for each pixel of the source image obtaining both rigid and non rigid displacements. On the contrary, Ref. [25] proposes a supervised deep learning based scheme for obtaining the displacements of the four corners of the source image. In addition, the proposed scheme is completely unsupervised, in contrast with other deep learning approaches like [27,28] that require ground truth deformations in order to be implemented. The requirement of ground truth deformations can be demanding and time-consuming while sometimes it can be quite difficult to be accomplished, as the available multitemporal images may have been collected from different aerial and satellite

sensors, with different spatial resolution and acquisition angles. This highlights the need for unsupervised methods such as our proposed method where the optimization strategy checks the similarity between the pair of images after warping the source image with the obtained deformation. One more key difference between our method and the schemes proposed in [27,28] is the use of spatial gradients for the regression of the deformation field. The definition of the deformation strategy is one of the most important elements for the learning based registration methods, with our framework proving its good performance on two different high resolution, urban and periurban regions. Finally, our work further contributes into the learning based registration methods by proposing a multistep approach that is able to refine in different steps the obtained by a deep learning scheme deformation.

One interesting topic when it comes to image registration is that of aligning images acquired by different sensors or with different spatial resolutions. Such a problem can be much more challenging since the images can be very diverse, including far more radiometric, textural and spatial differences [38]. As we have already mentioned, our method seems to cope with the local displacements more effectively, meaning that it can serve as a foundation for solving this demanding issue. Of course, it may be necessary to introduce more restrictions in order to assist the training process and create a model more robust to any kind of diversity between a given image pair. For example, an additional loss function could be established, focusing on the similarity of the image gradients [39]. Certainly, this is a vast research topic and requires much investigation and experiments, until a powerful model can be created.

## 5. Conclusions

In this study we tackle the challenging problem of image registration by employing a multistep framework based on fully convolutional networks. Experiments on two datasets of different resolutions demonstrated the high potentials of our method which managed to align the given image pairs successfully with overall, less than 3 pixels error. Our method is completely unsupervised while it does not require any preprocessing steps, providing an end-to-end framework. This means that the user is able to omit the process of creating ground truth deformations, something which usually requires much time and attention. Additionally, the proposed scheme is independent of the employed deep architecture while it also provides fast inference times. In the future, we plan to further improve our method by investigating additional formulations that can be adapted to the optimization problem, e.g., the regularization term in the loss function, introducing more restrictions or additional feature information. Additionally, we would like to evaluate our method on medium resolution datasets that contain mountainous areas with challenging elevation variations. Finally, we intend to combine the processes of image registration and change detection and evaluate the benefits of this synergy for both tasks.

**Author Contributions:** Conceptualization, M.P., S.C., K.K. and M.V.; methodology, M.P., M.V.; software, M.P.; validation, M.P., K.K. and M.V.; formal analysis, M.P., K.K. and M.V.; investigation, M.P. and M.V.; writing—original draft preparation, M.P. and M.V.; writing—review and editing, M.P., S.C., K.K. and M.V.; supervision, K.K. and M.V. All authors have read and agreed to the published version of the manuscript.

**Funding:** Part of this research was funded by the Research Committee of the National Technical University of Athens (Scholarship Grant).

**Institutional Review Board Statement:** Not applicable.

**Informed Consent Statement:** Not applicable.

**Data Availability Statement:** Not applicable.

**Conflicts of Interest:** The authors declare no conflict of interest.

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
