# Peer review of "Unsupervised Multistep Deformable Registration of Remote Sensing Imagery Based on Deep Learning"

_remotesensing, doi:10.3390/rs13071294_

Round 1

Reviewer 2 Report

This paper presents an image registration proposal based on Deep Learning. This proposal is an extension of a previous work by the authors [27], which is extended using a multi-step approach. The work is very interesting and clearly indicates the new contributions. It is also very well written.  

Nevertheless, authors should clarify the following points: 

  • A more detailed explanation of its improvement with respect to previous works [22], [24] and [25]. 
  • In the explanation of Table 1 in section 3.1, Roads and Fields areas are described in detail. However, the results for Building areas are not explained. The authors should also indicate what happens for T> 5 for these zones. 
  • The new proposal is compared with other proposal [27] that belongs to the same authors. I would like a comparison with previous proposal by other authors, such as [22]. I know that it is difficult to obtain the same dataset. But in proposal [22], it seems that results are more accuracy than presented in this paper. At least, a qualitative comparison would be performed. 

 On the other hand, section 2.3 should be moved to Section 3. 

Reviewer 3 Report

Quantitative and qualitative results are demonstrated the high potentials of the method, propose by the authors.

The developed methodology is evaluated in two different datasets as high-resolution dataset of the Attica, Greece and high resolution ISPRS Ikonos dataset.

The formulation of the authors can be integrated into convolutional architecture, providing at the same time fast inference performances.

The displacements are calculated with an iterative way, utilizing different time steps to refine and regress them. The presented method is based on the expression power of deep convolutional networks, and regression on the spatial gradients of the deformation and employing a 2D transformer layer to efficiently warp one image to the other.

The obtained results can be described as a fully unsupervised, deep learning based multi-step deformable registration scheme for aligning pairs of satellite imagery. These results can be used in processing image sensing images.

I have some reviewer notes:

1.Improve the discussion part. It will be good to compare your results with those from more authors. It will show in better way your contribution.

2.You can improve the conclusion part with information about the practical importance of your findings.

3.Also in 'Conclusion" part, it will be good to write what are the advantages of your results over known solutions in this study area.

Round 2

Reviewer 1 Report

The authors have answered my concerns.

Reviewer 2 Report

Authors have answered all my suggestions in an appropriate way.